# Industrial Water Pollution Discharge Taxes in China: A Multi-Sector Dynamic Analysis

**Xiaolin Guo [1], Mun Sing Ho [2]** 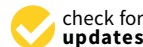**, Liangzhi You [3,4], Jing Cao [5,\*], Yu Fang [1], Taotao Tu [4] and Yang Hong [1,6,\*]**

[1] State Key Laboratory of Hydroscience and Engineering, Department of Hydraulic Engineering, Tsinghua University, Beijing 100084, China; guoxl15@mails.tsinghua.edu.cn (X.G.); fangyu516@mail.tsinghua.edu.cn (Y.F.)

[2] School of Engineering and Applied Sciences, Harvard University, Cambridge, MA 02138, USA; ho@rff.org

[3] Environment and Production Technology Division (EPTD), International Food Policy Research Institute, Washington, DC 20006, USA; l.you@cgiar.org

[4] College of Economics and Management, Huazhong Agricultural University, Wuhan 430070, China; tutaotao@mail.hzau.edu.cn

[5] School of Economics and Management, Tsinghua University, Beijing 100084, China

[6] Department of Civil Engineering and Environmental Science, University of Oklahoma, Norman, OK 73019, USA

[\*] Correspondence: caojing@sem.tsinghua.edu.cn (J.C.); hongyang@tsinghua.edu.cn (Y.H.); Tel.: +86-10-62787394 (Y.H.)

**Abstract:** We explore how water pollution policy reforms in China could reduce industrial wastewater pollution with minimum adverse impact on GDP growth. We use a multi-sector dynamic Computable General Equilibrium (CGE) model, jointly developed by Harvard University and Tsinghua University, to examine the long-term impact of pollution taxes. A firm-level dataset of wastewater and COD discharge is compiled and aggregated to provide COD-intensities for 22 industrial sectors. We simulated the impact of 4 different sets of Pigovian taxes on the output of these industrial sectors, where the tax rate depends on the COD-output intensity. In the baseline low rate of COD tax, COD discharge is projected to rise from 36 million tons in 2018 to 48 million in 2030, while GDP grows at 6.9% per year. We find that raising the COD tax by 8 times will lower COD discharge by 1.6% by 2030, while a high 20-times tax will cut it by 4.0%. The most COD-intensive sectors—textile goods, apparel, and food products—have the biggest reduction in output and emissions. The additional tax revenue is recycled by cutting existing taxes, including taxes on profits, leading to higher investment. This shift from consumption to investment leads to a slightly higher GDP over time.

**Keywords:** water pollution policy; CGE model; industrial COD discharge; pollution discharge tax; China

## 1. Introduction

Population growth and rapid economic development in recent decades have dramatically aggravated water pollution and water withdrawal in China, while climate change is resulting in an increasingly uncertain water supply and places additional pressure on constrained freshwater resources [1–3]. Thus, a series of policies have been proposed by the Chinese government to address the challenges posed by water shortage and water quality degradation. Before 2015, China's Wastewater Discharge Fees were based on treatment costs under the "Regulations on Collection, Use & Management of Wastewater Charges". The low discharge fees led to high wastewater pollution; furthermore, low water prices did not encourage companies to recycle water, reduce usage and reduce

wastewater discharge. China's total wastewater discharge has steadily risen from 41.5 billion tons in 2000 to 73.5 billion tons in 2015 [4].

In recognition of these problems, China finally reformed and reinforced its environmental regulations with respect to water. First, the new Environmental Protection Law (EPL), enacted in January 2015, has become the most progressive and stringent law in the history of environmental protection in China [5]. Second, the State Council issued the 'Water Pollution Prevention and Control Action Plan' (the 'Ten-Point Water Plan') on 16 April 2015. During the first two years of the new EPL's implementation, companies associated with wastewater (including both discharge and treatment plants) were still the major polluters with excessive discharge [6]. The 2016 State of Environment Report from the Ministry of Ecology and the Environment (formerly the Ministry of Environment Protection, MEP) indicated that 2016 was a decisive year for the foundation of a good-quality environment. The report noted significant improvements in several aspects of water management, but also emphasized that much progress is still needed to win the "war on pollution". For example, the surface water quality of individual rivers is not close to reaching the Ten-Point Water Plan target of having 70% of surface water meeting Grade III (The Ministry classify surface water quality into Grades I–V. Grades I through III are "suitable for domestic use," while Grades IV and V are "not for human contact," and "suitable for agriculture use") or better by 2020, despite there being a slight overall improvement compared to 2015 [7].

Industrial wastewater, as one of the main sources of wastewater discharge, is reported to have been approximately 20–25 billion tons per year since 2000 and has remained a big concern for the government [8]. "Industrial" refers to the mining, manufacturing and utility sectors. The real number may be even higher, due to underreporting and mismatch in both water quality standard and wastewater standard. Previous studies have also pointed out that many newly established industry plants do not control pollution as effectively as factories in other parts of the world [2,9].

More recently, the Chinese government has emphasized the role that markets could play in allocating resources, and attempted to reform the tax system, launching a comprehensive reform to convert resource fees to resource taxes, with Hebei as the pilot province [10]. On 9 May 2016, the 'Interim Measures for the Pilot Reform of Water Resource Tax' ("Water Tax Interim Measures") for Hebei province were issued by the Ministry of Finance (MoF), the State Administration of Taxation (SAT) and the Ministry of Water Resources (MWR) [11]. The water tax (Environmental Protection Tax Law) has been in force since 1 January 2018. According to the SAT, there are three main differences between fees and taxes: first, the main tax-collection bodies are the tax authorities, customs and fiscal authorities representing the state, while fees are collected by sector administrations (e.g., water resource fees are usually collected by water administrations). Second, tax rates are fixed, while fees are determined based on treatment costs. Third, revenues collected from fees are used for water conservation, while revenues from taxes can be used by the State for any purpose. Before the transition, China collected water resource fees, which were specifically used for water conservation. However, there are some problems regarding the fee, including the lack of awareness of paying fees and the lack of mandatory collection measures, among others [11].

The switch from fees to taxes is a big policy change but has attracted little analysis to date. The objective of this study is to evaluate the impacts of different tax policies on the economy and to assess the effectiveness of different policy options. The polluter pays principle is first recognized as an internationally agreed principle for environmental policy by the Organization for Economic Cooperation and Development in 1972 [12,13]. It is defined as the principle whereby the polluter should bear the expenses of carrying out the pollution prevention and control measures decided by public authorities to ensure that the environment is in an acceptable state [14]. We focus on taxes on industry output where the tax rate is based on the amount of pollution generated in producing that output—what economists call Pigovian taxes. The purpose of this tax is to inject price signals into the economic system so that consumers take into account the environmental damage caused by that good when deciding how much to buy [13]. Governments may also consider a direct tax on pollution

emissions such as COD discharge; such a tax is efficient, but may be difficult to apply to all emissions. Under the current system, the pollution tax only applied to large enterprises.

We focus on Chemical Oxygen Demand (COD), a measure of the oxygen needed to dissolve organic matter in wastewater, since it is regarded as the most important form of water pollution in China [8]. We use a national dataset of firms to establish a detailed picture of water use and discharge by different industries and characteristics of firms, and then develop a CGE model to estimate the environmental impacts (freshwater use and COD discharge) and economic impacts (GDP and output) of four tax policies by considering the intensity of COD discharge. Here We use the term "COD discharge" to refer to the measure of organic and inorganic discharge that contributes to the COD index, this is also the term used in the China statistical books. The baseline policy is the current level of water taxes, and we consider 3 alternative policies where we increase the baseline rate by 4 times, 8 times and 20 times, matching the highest provincial tax rates today. We also consider a hybrid policy of taxing the more polluting sectors at a higher rate, and others at lower rates.

We find that a policy that taxes the output of each industry at rates based on COD discharge intensities is effective in reducing water pollution while having only a modest negative impact on consumption. When the new pollution tax revenue is used to cut other existing taxes, there could even be a positive impact on GDP. The higher prices for dirty goods encourage conservation and substitution to other inputs resulting in lower pollution.

We analyze these policies using a class of models in economics known as Computable General Equilibrium (CGE) models. Over the past 25 years, CGE models have become a standard tool of empirical economic analysis and have been widely applied to assess the impact of environmental policies, such as those reviewed in [15]. CGE models account for interindustry transactions, e.g., how much electricity is bought by the steel industry, how much steel is bought by vehicle manufacturers, and how many vehicles are bought by consumers? On the supply side, there are production functions that give the inputs required to produce a unit of each industry's output as a function of input prices and gives information on the pollution emissions by that industry. On the demand side, there are consumption functions that describe how households allocate their total spending to the different goods based on incomes and prices. These models are "equilibrium" in that they solve for prices that balance supply and demand for each good and for capital and labor inputs; they are "general" in that they cover the entire economy (In contrast, a "partial equilibrium" approach would consider only one market in great detail, e.g., a model of the electricity market. The CGE approach has fewer technical details but trace the impact of higher electricity taxes on all sectors and households). In the CGE model, a new policy such as a COD tax would change the costs, and therefore the prices, of different goods to different extents. The new set of relative prices would change consumption behavior, and thus lead to a new vector of industry output.

In recent years, improvements in model specification, data availability, and computer technology have improved the payoffs and reduced the costs of policy analysis based on CGE models, paving the way for their widespread use by policy analysts throughout the world. Recent studies of water resource policies have focused on agriculture and used CGE models, for example, allocation of water rights [16], water pricing, water withdrawal limits and agricultural water consumption [17–19]. Some other studies used CGE models to assess the impact of industry water policies; Fang, G. demonstrated that an increase in discharge fees would have a negative impact on economic output but lead to a less environmentally damaging industrial structure, contributing to sustainable development [1]. The simulations in Wen Chen of Hunan province indicate that a water pollution tax has a smaller negative effect on GDP in the short term than in long term, and that the manufacturing sector would suffer the most in short term while the wool textile dying industry suffers the most in the long term [20]. Yijun Yuan also used an environmental CGE model to show that the water pollution tax would reduce the COD emissions significantly and lead to a decline in GDP and outputs of most industrial sectors [21]. Our results for output taxes are not directly comparable with the previous research which examine a fee or tax on the discharge. They are consistent with them in that we also

show a reduction in pollution and a reduction in output of many industrial sectors. However, we also include the revenue recycling effect and the impact on investment and future GDP growth.

These studies of industrial wastewater applied static models to simulate the economic and environmental impacts and do not discuss the dynamic aspects of policies. Here we employ a dynamic multi-sector model of the Chinese economy to assess the impact of the new water pollution tax and possible extensions of the tax. This growth model of China has been used in many earlier analyses of air pollution policies and carbon price policies [9,15]. In this paper, we develop new parameters for water use and emissions using enterprise survey data and include them in this CGE model. These parameters are described in Section 2, and the economic model is described in Section 2.3.

## 2. Materials and Methods

### 2.1. Input Dataset

A key input into the model is a Social Accounting Matrix (SAM), which we constructed in 2014. This traces the flow of monetary transactions (sales, transfers, taxes) between industries, households, government and the "rest of the world". The main component of the SAM is the input–output (IO) matrix, which gives the whole vector of inputs purchased by each of our 33 industries. The output from each industry is combined with imports to form the total supply of a given good. The total supply of each good (say, steel) is completely accounted for; it equals the sum of all purchases of steel—purchases by other industries as intermediate input and purchases by "final demand" (households, government, investment and exports). Our 2014 input–output table is derived from the official 2012 benchmark IO table. The benchmark IO table for 2012 (NBS 2016) is derived from detailed enterprise data by the NBS, which Cao Jing use to extrapolate the 2014 IO table using industry output, value added and trade data for 2014 [22]. Of the 33 sectors and commodities identified, 22 are industrial sectors (mining, manufacturing and utilities). The SAM includes information on payments to workers and capital owners, the expenditures by households on each commodity, the taxes paid and transfers received by households, the expenditure on incomes of the government.

We supplemented the SAM by collecting industry-specific national data on fresh water use, wastewater discharge and "COD discharge" from the following four sources: (1) firm-level pollution data from the Ministry of Environment Protection (MEP) of China [2,23]; (2) the Industry Enterprise Dataset from the National Bureau of Statistics of China (2008) [2,23]; (3) China Statistical Yearbook 2015 (http://www.stats.gov.cn/tjsj/ndsj/2015/indexch.htm); (4) China Environmental Statistics Annual Report 2014 (http://www.stats.gov.cn/ztjc/ztsj/hjtjzl/). We use the term "COD discharge" to refer to the measure of organic and inorganic discharge that contributes to the COD index, this is also the term used in the China statistical books.

The major source of industry-level information for our study is the Environmental Survey and Report made by the MEP, which provides extensive information on pollutant emissions, including freshwater use, wastewater discharge and COD discharge, for a total of 80,626 firms nationwide (The Environmental Survey and Report has compiled data on industrial firms since the 1980s but has only recently made the data available to researchers). It is the most comprehensive source for studies of industry pollution [2,23]. This dataset has several implausible records (extreme values or extreme jumps), and we excluded firms if: (1) output value is zero or missing; (2) the change in freshwater use intensity between two years exceeds 500%. We tested alternative criteria, e.g., also excluding jumps in wastewater intensity exceeding 500% and jumps exceeding 400%. These alternatives changed our simulated policy impacts by about 15%. Given this modest impact, we chose the simplest criteria, and the resulting pollution dataset contains records for 42,415 firms with data for 2005 to 2009.

The National Bureau of Statistics (NBS) Industry Enterprise database contains the firm name, city code, date of establishment, industry classification, output, sales, and so on. We link the NBS Industry Enterprise Dataset to the MEP pollution dataset according to firm name and location, so that we get

the industry classification and output. Output is deflated using producer price indexes from the China Statistical Yearbook of 2015.

We thus developed a national dataset on industrial pollution with firm-level information on industry classification, industry code, output, resource inputs, and pollutant-discharge levels. We aggregated the firm data into 22 sectors (Table 1), which is consistent with the commodity and industry accounts in our CGE model. The economics literature shows how enterprise performance differs based on characteristics such as size, location, ownership status and exposure to competition [24]. In our model we ignore these differences within a given industry and focus on differences in the averages between industries. Estimating the differences in production processes and pollution control behavior is a challenging task that we have to defer to future work.

*2.2. Water Use Intensity, COD Discharge Rate and Concentration*

The volume of water use and discharge vary considerably among sectors. We begin by defining the intensities of freshwater use ($I_{j,t}^{FW}$), COD discharge ($I_{j,t}^{COD}$) and COD concentration ($S_{j,t}^{COD}$) of 22 industrial sectors ($j$) as:

$$I_{j,2008}^{FW} = \frac{FW_{j,2008}}{QI_{j,2008}}, \tag{1}$$

$$I_{j,2008}^{COD} = \frac{COD_{j,2008}}{QI_{j,2008}}, \tag{2}$$

$$S_{j,2008}^{COD} = \frac{COD_{j,2008}}{WW_{j,2008}}, \tag{3}$$

where $I_{j,t}^{FW}$ (ton/bilion yuan) is the intensity of freshwater use, $FW_{j,t}$ (ton) is the quantity of freshwater use, $QI_{j,t}$ (billion yuan) is the quantity of industry output in constant 2014 yuan, $COD_{j,t}$ (ton) is the COD discharge, $WW_{j,t}$ (ton) is the wastewater discharge, and subscript $j$ denotes industry and $t$ years. $S_{j,t}^{COD}$ is the organic and inorganic pollutant concentration in wastewater. We first obtain a preliminary estimate of the industry $j$ intensities using the average of the firms in $j$ in our 2008 national industry dataset.

We then calibrate the freshwater intensity to the total industrial freshwater use from the China Statistical Yearbook (2015, Table 8-12, "Water Supply and Water Use") which gives subtotals for agriculture, industry (mining, manufacturing, utilities) and commercial/household consumption. "Industrial Output" is taken from the input–output table. The calibrated 2014 intensities are given by:

$$I_{j,2014}^{FW} = I_{j,2008}^{FW} \cdot \frac{QI_{j,2008}}{QI_{j,2014}} \cdot \frac{FW_{2014}^{IND}}{FW_{2008}^{IND}} \tag{4}$$

where $FW_t^{IND}$ is the industrial freshwater use from the China Statistical Yearbook, and real output $QI_{j,t}$ is obtained by deflating the nominal output $VQI_{j,2014}$ from the IO table by the producer price indexes from the same China Statistical Yearbook.

We also calibrate the COD discharge coefficients using the national totals in China Environmental Statistics Annual Report 2014 (Table 2-1, "Discharge of Wastewater and Major Pollutants in China"):

$$I_{j,2014}^{COD} = I_{j,2008}^{COD} \cdot \frac{QI_{j,2008}}{QI_{j,2014}} \cdot \frac{\sum_j COD_{j,2014}}{\sum_j COD_{j,2008}} \tag{5}$$

where $COD_{j,2014}$ is the industrial COD from the China Environmental Statistics Annual Report. A parallel $I_{j,2014}^{WW}$ coefficient is defined for wastewater discharge. In this study, we fix these intensities for all the simulations.

The COD intensity and COD concentration for 22 industrial sectors are given in Table 1. COD intensity is highest for Non-energy mining (1.14 kg/1000 yuan), Textile goods (0.93), Other

mining industries, and Apparel-leather (0.66), while the lowest is petroleum processing and coking (0.03). The range of concentrations of the discharge is much smaller than the intensities, from 0.57 for mining and food products to 0.2 for instruments. Total estimated COD discharge for each industry in 2014 is given in the last column of Table 1. The biggest emitters are Textile goods (4.02 million tons), Chemical (3.90 million tons), Machinery (3.47 million tons) and Food products and tobacco processing (2.96 million tons).

**Table 1.** The values of coefficients in the discharge equations used to estimate COD discharge in 2014.

| Sector | COD Concentration $S_{j,t}^{COD}$ (10$^{-3}$) | COD Intensity $I_{j,t}^{COD}$ (kg/1000 yuan) | Wastewater Discharge (billion ton) | COD Discharge (million ton) |
|---|---|---|---|---|
| Coal mining | 0.57 | 0.64 | 2.46 | 1.40 |
| Oil mining | 0.57 | 0.89 | 1.50 | 0.85 |
| Natural gas mining | 0.57 | 0.89 | 0.35 | 0.20 |
| Non-energy mining | 0.56 | 1.14 | 3.97 | 2.23 |
| Food products and tobacco processing | 0.57 | 0.28 | 5.23 | 2.96 |
| Textile goods | 0.53 | 0.93 | 7.54 | 4.02 |
| Wearing apparel, leather, furs, down and related products | 0.53 | 0.66 | 4.54 | 2.38 |
| Sawmills and furniture | 0.50 | 0.24 | 1.12 | 0.56 |
| Paper and products, printing and record medium reproduction | 0.45 | 0.61 | 5.18 | 2.32 |
| Petroleum processing and coking | 0.42 | 0.03 | 0.34 | 0.14 |
| Chemical | 0.39 | 0.27 | 10.01 | 3.90 |
| Nonmetal mineral products | 0.41 | 0.35 | 5.17 | 2.12 |
| Metals smelting and pressing | 0.39 | 0.10 | 3.10 | 1.20 |
| Metal products | 0.39 | 0.23 | 2.47 | 0.95 |
| Machinery | 0.39 | 0.41 | 8.97 | 3.47 |
| Transportation equipment | 0.39 | 0.11 | 2.29 | 0.88 |
| Electrical machinery | 0.39 | 0.09 | 1.39 | 0.54 |
| Communication equipment, computer, Electronic | 0.39 | 0.08 | 1.73 | 0.67 |
| Instruments | 0.20 | 0.28 | 0.52 | 0.10 |
| Other manufacturing products | 0.22 | 0.20 | 0.73 | 0.16 |
| Electricity, steam and hot water production and supply | 0.20 | 0.20 | 3.80 | 0.75 |
| Gas production and supply | 0.20 | 0.24 | 0.59 | 0.12 |
| National | 0.42 | 0.41 | 73.0 | 31.90 |

## 2.3. Dynamic CGE Model

The economic and environment model used here is an updated version of one used to study the co-benefits of the 11th Five-Year Plan SO$_2$ policies and carbon price policies as described in Garbaccio, Nielsen and Ho [15,25]. It is a dynamic recursive CGE model with 33 sectors, including 22 manufacturing and mining industries (see Table 1 for the list of sectors). And dynamic recursive means that savings, and hence, investment, is driven by a simple fixed savings rate. This is in contrast to inter-temporal equilibrium models with endogenous savings rate. Economic growth is driven by investment, labor force growth and total factor productivity (TFP) growth. The exogenous drivers are the private savings rate, dividend rate, TFP growth, capital quality growth and effective labor growth. Our assumption is that the saving rate will fall towards levels observed for rich countries, beginning at the observed 38.9% for 2014 and falling to 30.8% in 2020 and 22.6% in 2030 (see also, [26]). National private savings is household savings plus the retained earnings of enterprises. The share of retained earnings is assumed to fall, and dividend payouts to rise to reflect the diminishing role of state enterprises in the economy. The dividend rate, i.e., the share not used for retained earnings, was 41.7% in 2014, and we project it to rise to 53% by 2020. It should be pointed out that national savings and investment in China includes capital investments such as roads and other public infrastructure; items that are excluded from the "gross fixed private investment" item in the National Accounts of most other countries.

Effective labor supply is not a simple number of working age people, but an index that depends on the age and educational attainment constructed using detailed labor data given by Wu [27].

We combine 2010 labor data with population projections by age group from the Population Division of the Department of Economic and Social Affairs of the United Nations Secretariat (The demographic projections are from World Population Prospects: the 2015 Revision downloaded from the U.N. web site, https://esa.un.org/unpd/wpp/). The composition of the work force changes over time, trending towards a higher proportion of educated workers, later retirement and an older average age. In addition to these effects, we project labor supply to grow at −0.5% per year during 2014–2030. Finally, we calibrate TFP growth so that the base case GDP growth is 5.2% during 2018–2030, close to the mainstream projections by various authors ( For example, World Bank and DRC (2013), before the recent deceleration of GDP growth, projected a 5.4% rate for 2020–2030, while IEA (2017) projects 5.8% (2016–2025) and 3.7% (2025–2040) [28]. The main features of the model are summarized in Appendix A, a more complete description of the model and parameters is given in Cao and Ho (2018) [22]. Production function elasticities are taken from the Global Trade Analysis Project (GTAP) model and projections of input intensities are developed in comparison to advanced economies. Notice that the GTAP is a global network of researchers and policy makers conducting quantitative analysis of international policy issues and is described in www.gtap.agecon.purdue.edu.

As has been emphasized by many researchers on environmental taxes, the method of recycling pollution tax revenues has an important impact on the net economic cost of such a tax [29]. Here the water pollution tax is offset by cuts in the value-added tax (VAT) and capital income tax.

Our approach to estimating the impact of the policies is to first simulate a base case with the economic model where we assume that the tax structure of the base year 2018 is maintained for the projection period. That is, there are no new taxes in the base case path. This model is solved annually for a set of prices that clears all markets. The model is then solved again for four policy scenarios, which are discussed next.

*2.4. Water Pollution Tax Scenarios*

Before the 2015 reform that replaced resource fees with taxes, the discharge standards consisted of three categories: "Integrated Wastewater Discharge Standard (GB8978-1996)", "Discharge Standard of Pollutants for Municipal Wastewater Treatment Plant (GB18918-2002)", and discharge standards of industrial pollutants [30]. The pollution discharge fee is 0.7 yuan per unit of water pollution [1]. After the new regulations, the "pollutant discharge fee" will be abolished. According to the Finance Ministry, local governments will have more autonomy to set a range for the taxes—from 1.4 yuan to 14 yuan for each unit of water pollution emitted—to reflect the different regional environmental and economic conditions (news release by the Ministry of Ecology and Environment, 11 January 2018).

We do not implement a direct emission tax here, but instead impose a tax on industry output based on the COD discharge intensity. We calculate the total COD discharge taxes payable by each industry in the base year 2018 and divide that by the value of industry output in 2018 to give the average COD tax rate. This tax rate is applied to total output in each of the 22 industrial sectors, in contrast to the actual fees, which are only applied to large enterprises. The tax rate on output is thus proportional to the COD intensity given in the second column in Table 1. The estimates of the tax rates for each province is given in Figure 1 (News release by the Ministry of Ecology and Environment, 11 July 2018, at: http://kjs.mee.gov.cn/hjbhbz/dfhjbhbzba/201807/t20180711_446455.shtml).

We simulate four scenarios with two dimensions: one relates to the level of tax on COD discharge, and the second relates to the industries covered by the tax. The tax rates are represented by the horizontal lines in Figure 1. The baseline incorporates the current tax which averages to 0.7 yuan/kg of COD discharge. In the 3 alternative policy scenarios, we raise the COD tax on all the industrial sectors by "4-times", "8-times", and "20-times" the baseline level, or 2.8, 5.6 and 14 yuan/kg, respectively. These scenarios are chosen to reflect the wide range of 1.4 to 14 yuan/kg COD tax allowed by the Ministry of Finance. In the fourth policy case, the Hybrid scenario, a tax of 14 yuan is placed on sectors with high COD concentrations ($S_{j,t}^{COD}$) and a tax of 2.8 yuan on the low concentration

ones. The dividing line between the high and low groups is the average industrial COD discharge concentration (i.e., averaged over the industries in Table 1). These new policy taxes all start in 2018.

To simplify the comparison of welfare across different tax cases we maintain the path of real government purchases in each policy simulation at the baseline growth path. The baseline path is determined by assumptions regarding tax rates, overall economic growth and deficits. To maintain comparable aggregate real purchases, we implement a revenue-neutral change by using the new revenues raised by higher water taxes to cut other existing taxes. This allows us to focus on comparing the consumption and investment changes between the policy and baseline without any change in the quantity of public goods.

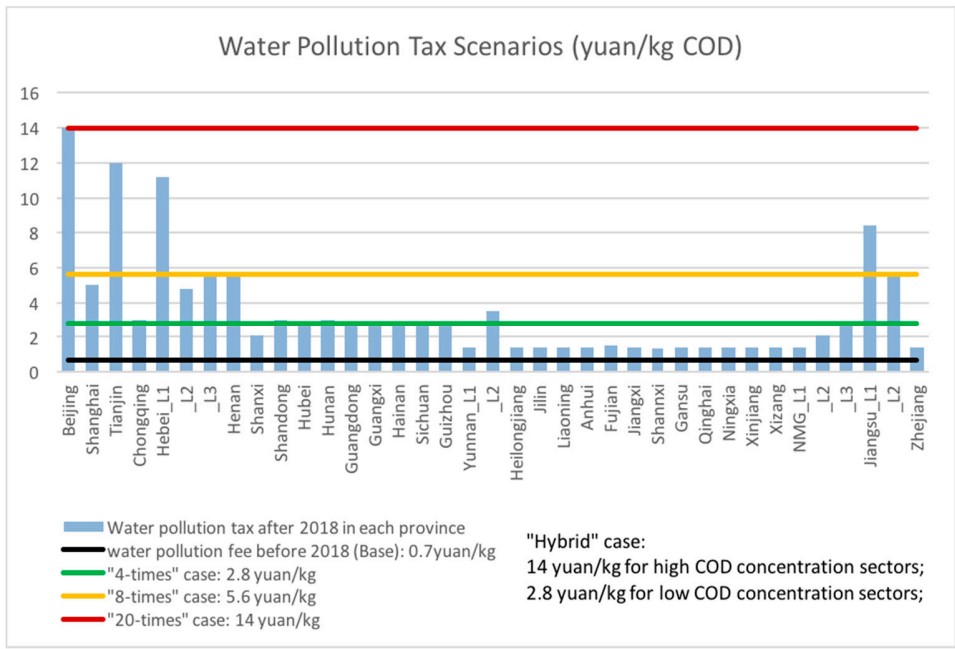

**Figure 1.** Four water pollution tax policy scenarios of China. Here the high discharge sectors are Coal mining; Oil mining; Natural gas mining; Non-energy mining; Food products and tobacco processing; Textile goods; Wearing apparel, leather, furs, down and related products; Paper and products, printing and record medium reproduction.

## 3. Simulation Results

We begin by describing the base case in Section 3.1 to give an idea of the projected transformation of the rapidly growing economy. Section 3.2 discusses the impacts of pollution-discharge taxes in the three alternative tax policy scenarios.

### 3.1. Baseline

The main economic and water variables in the baseline are given in Table 2 for 2018 and 2030. As we noted, aggregate GDP growth is mainly determined by the projections of exogenous variables—labor growth, productivity growth and savings rate. Consumption refers to private household consumption, and Investment corresponds to the concept in the Chinese National Accounts which includes both private and public investment. We do not include government consumption in Table 2, since it is held fixed by construction. The change in composition of the economy is affected by changes in consumption/investment ratio, by income effects on consumption, and by changes in technology. The population is projected to rise modestly from 1393 to 1416 million between 2018 and 2030, and real GDP grows at an average of 5.2% per year. In the baseline, total wastewater grows from 83 billion ton in 2018 to 111 in 2030, while total COD discharge increases from 36 to 48 billion kg (growing at 2.8% per year during 2018–2030). The major COD discharge sources in 2018 are Textile

goods, Wearing apparel, Chemical, Machinery, Food products and tobacco processing, and their COD discharges will grow at 1.1%, 5.9%, 3.1%, 2.0% and 2.7% per year during 2018–2030, respectively. These slow growth rates reflect the overall shift from manufacturing to services in the economy.

**Table 2.** The overall impacts of different water pollution tax scenarios compared with the baseline.

| Variable | 2018 | | | | | 2030 | | | | |
|---|---|---|---|---|---|---|---|---|---|---|
| | Base | 4 Times | 8 Times | 20 Times | Hybrid | Base | 4 Times | 8 Times | 20 Times | Hybrid |
| | | % Change from Base Case | | | | | % Change from Base Case | | | |
| Population (million) | 1393 | | | | | 1416 | | | | |
| Real GDP (billion 2014 yuan) | 82,316 | 0.02 | 0.04 | 0.09 | 0.06 | 150,957 | 0.06 | 0.13 | 0.32 | 0.42 |
| Consumption (billion 2014 yuan) | 34,818 | −0.02 | −0.04 | −0.23 | −0.12 | 75,507 | 0.06 | 0.13 | 0.32 | 0.24 |
| Investment (billion 2014 yuan) | 35,425 | 0.07 | 0.16 | 0.41 | 0.45 | 56,593 | 0.16 | 0.36 | 0.89 | 1.09 |
| Wastewater Use (billion ton) | 82.7 | −0.37 | −0.85 | −2.20 | −1.91 | 111.1 | −0.61 | −1.38 | −3.48 | −2.66 |
| COD Discharge (billion kg) | 36.0 | −0.44 | −1.00 | −2.58 | −2.41 | 48.0 | −0.72 | −1.62 | −4.02 | −3.42 |

### 3.2. Impact of Taxes

The results of the four water pollution tax scenarios for the main variables are given in Table 2 for 2018 and 2030 and the impact on total consumption over time is plotted in Figure 2. The initial impact on the overall economy is very small; in the 4-times scenario there is a 0.02% reduction in consumption and a 0.02% increase in GDP due to a small improvement in the terms of trade that reduced exports and allows higher investment. Recall that we cut existing taxes so that we have a revenue-neutral policy that keeps real government spending unchanged. The cut in capital taxes allows a higher rate of investment, accumulating into a higher stock of capital, which results in a slightly higher GDP in the later years, as shown in Figure 2. GDP in 2030 is higher by 0.06%. The changes in GDP and consumption in the 8-times and 20-times cases have a similar pattern as in the 4-times case, a small initial reduction in consumption and an increase in investment that eventually leads to a positive effect in 2030 GDP (see GDP row in Table 2). If one is using a foresighted model where the household maximizes a discounted stream of consumption, then this utility function could form the basis of a calculation of the equivalent variation of the policy effect. However, this is a recursive (myopic) model and we can only consider the annual consumption effects without defining a present value using some arbitrary discount rate.

This shift from consumption to investment contributes to a change in the production structure of the economy, since investment demand is concentrated on the output of Construction, Machinery, Electrical machinery, Transportation equipment and other heavy sectors. Capital and labor resources are thus shifted out of Agriculture, Food manufacturing, Wearing apparel, and Real estate to these heavy industries due to the final demand effect.

Another source of change in the composition of the economy is the industry price effect. The tax on COD discharge raises the costs to COD-intensive sectors such as Coal and oil mining, Food products and tobacco processing, and Textile goods. The general equilibrium impact on industry prices is shown in Figure 3 for the four scenarios for 2030; this includes the impact of the Pigovian tax and the impact of changes in the relative demand for consumption versus investment. For Textiles, the price rises by 1.0% in the 4-times case and by 5.7% in the 20-times case, while the price of paper rises by 3.1% in the 20-times case. At the other end of the impacts, the price of Commerce falls by 1.4% and Finance by 0.9% by 2030 in the 20-times case. The relative magnitudes of the price changes are similar between the 4-, 8- and 20-times cases—a big rise in the COD-intensive sectors and a decrease in services prices.

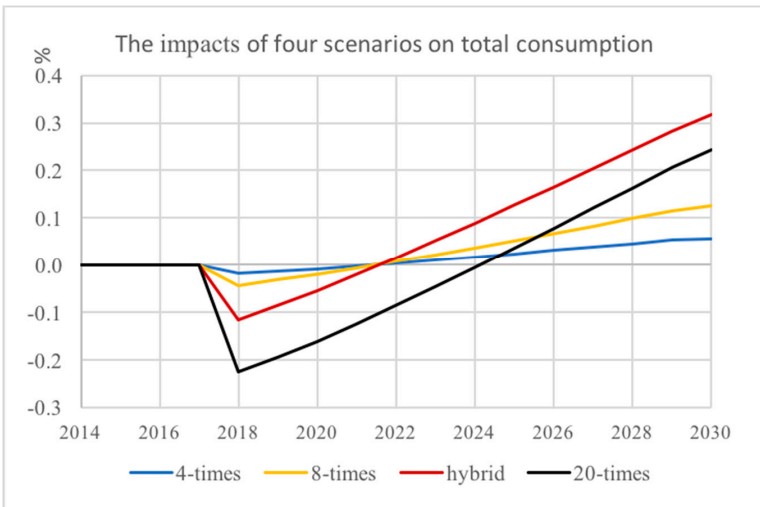

**Figure 2.** The impacts of four water pollution tax policy scenarios on total consumption (base year price).

The price changes in the Hybrid scenario (the plain bar in Figure 3) diverges somewhat from those changes. In the Hybrid case, the prices of the COD-intensive sectors also rise significantly, close to the changes in the 20-times case given the high tax placed on them. However, for some of the low COD-intensity industries with the low COD taxes, prices fall due to general equilibrium effects with capital and labor moving out of the COD-intensive sectors and into the low COD sectors. There is also a reduction in the relative prices of the services sectors.

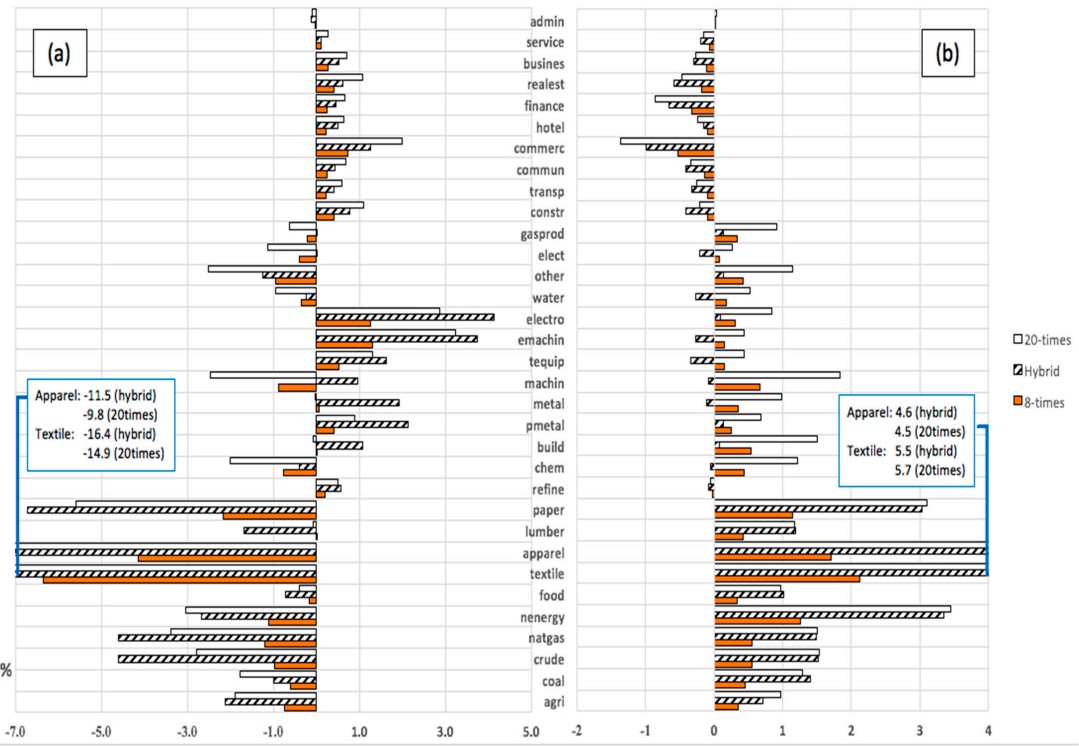

**Figure 3.** Impact of water pollution taxes on (**a**) industry output, and (**b**) industry prices in 2030 (percent change from base case). All abbreviations are shown in Table A1.

These price changes cut down on the demand for COD-intensive goods, and their output falls as shown in panel (a) in Figure 3 for 2030. Textile output falls by 14.9%, and paper by 5.6%,

while Commerce output increases by 2.0% under the '20-times' policy scenario. These output changes lead to a fall in aggregate COD discharge and wastewater discharge (last two rows of Table 2). The impact on total COD discharge is plotted for every year in Figure 4. In the first year, COD discharge is reduced by 160 million kg (0.44%), 360 million kg (1.00%), 870 million kg (2.41%) and 930 million kg (2.58%) for the "4-times", "8-times", "Hybrid" and "20-times" scenarios, respectively. Over time, the reductions get bigger, and by 2030, the discharge falls by 343 million kg (0.72%), 775 million kg (1.62%), 1641 million kg (3.42%) and 1930 million kg (4.02%) in these four cases. Figure 5 gives the COD-discharge intensity (million kg/billion yuan of GDP) over time. Since GDP rises slightly over time in the tax scenarios, the fall in COD-discharge intensity diminishes slightly over time too.

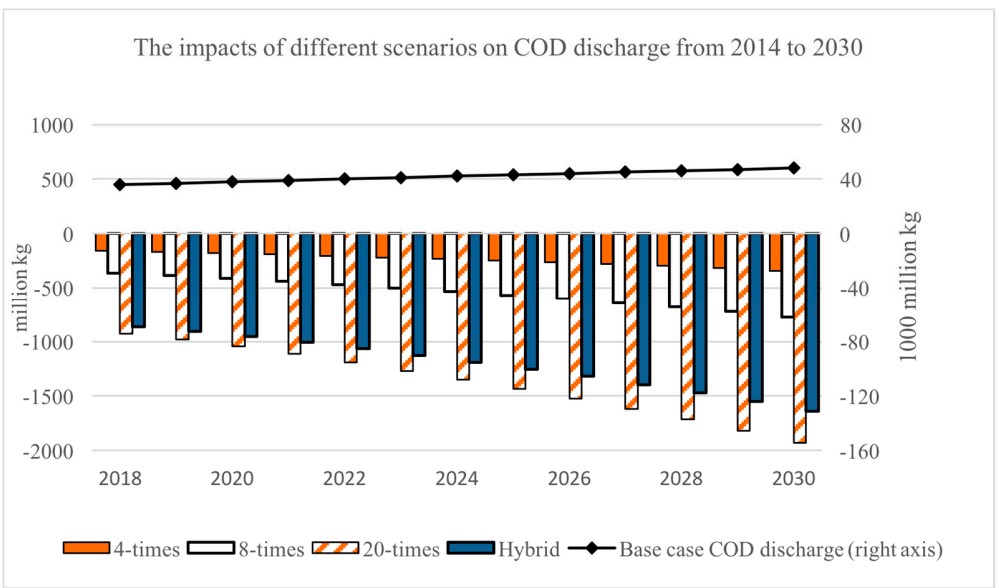

**Figure 4.** The impacts of different scenarios on COD discharge from 2018 to 2030. Negative values in the discharge change (left axis) mean that the total COD discharge is less than the total discharge in baseline (black line, right axis).

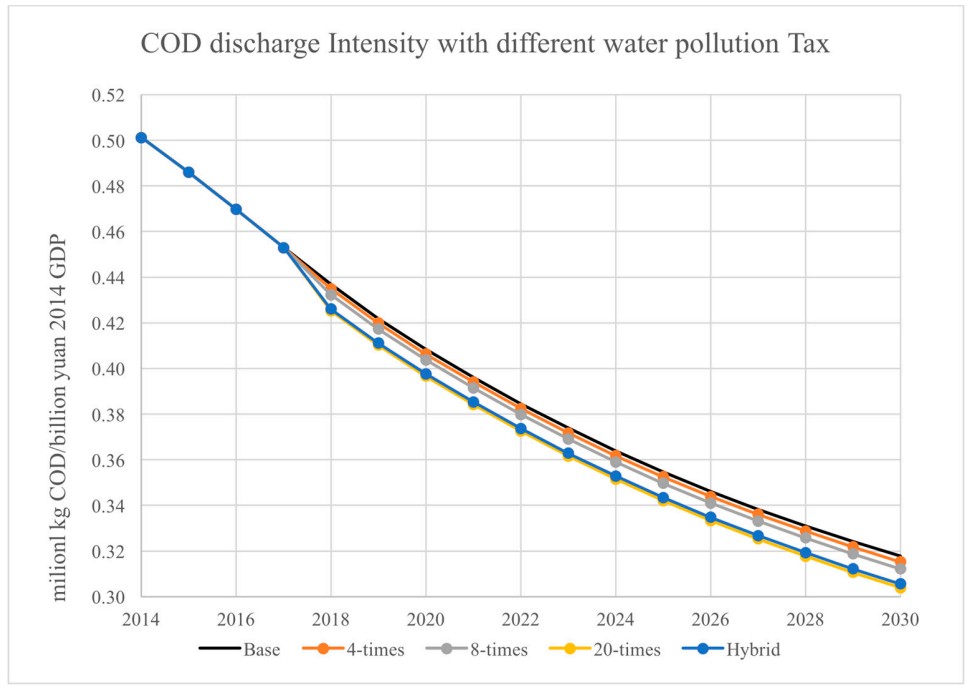

**Figure 5.** Time path of COD discharge Intensity at different tax levels.

The percent change in COD discharge in each of the 22 industrial sectors in the 4 policy cases for 2030 is shown in Figure 6. As we noted from Table 1, Textile, Apparel and Paper are the sectors with the highest COD intensities (kg/yuan), and thus face the highest tax rates under our output-based tax system. These three sectors have the largest reductions for each scenario, falling by 3.2%, 2.2% and 1.1% in the mild 4-times scenario. The 2030 COD reduction in Textile is 7.1% in the 8-times case, 16.6% in the 20-times and 17.9% in the Hybrid case. Most of the other industrial sectors see a small reduction, but 4 of them see an increase in COD discharge (Electronics, Electrical machinery, Transportation equipment and Primary metals). The higher COD discharge is due to the expansion of output because of the change in the composition of the economy due to the shift from Consumption to Investment.

We should note that the COD intensity is high for Oil mining, Gas mining and Non-energy mining, about the same level as Textile, and so they face a high COD tax too. However, as shown in Figure 3, the rise in their output price is not as large as the increase for Textile. These sectors use land (strictly speaking, natural resources in the land) as an input, unlike the manufacturing sectors, and the owners of the fixed resource bear a part of this new output-based COD tax. That is, as demand for the expensive COD-intensive goods fall, the other factors like capital and labor can move out to the non-COD-intensive sectors and not suffer a big change in returns; however, resources have no alternative use, and their return falls. The net change in their output price is thus dampened compared to the manufacturing sectors without a fixed land input. Even the most COD-intensive sector, Non-energy mining, sees an increase in 2030 output price of only 3.4%, compared to 5.7% in Textile goods. A smaller price change leads to a smaller reduction in sales and smaller reduction in COD discharge from these mining industries.

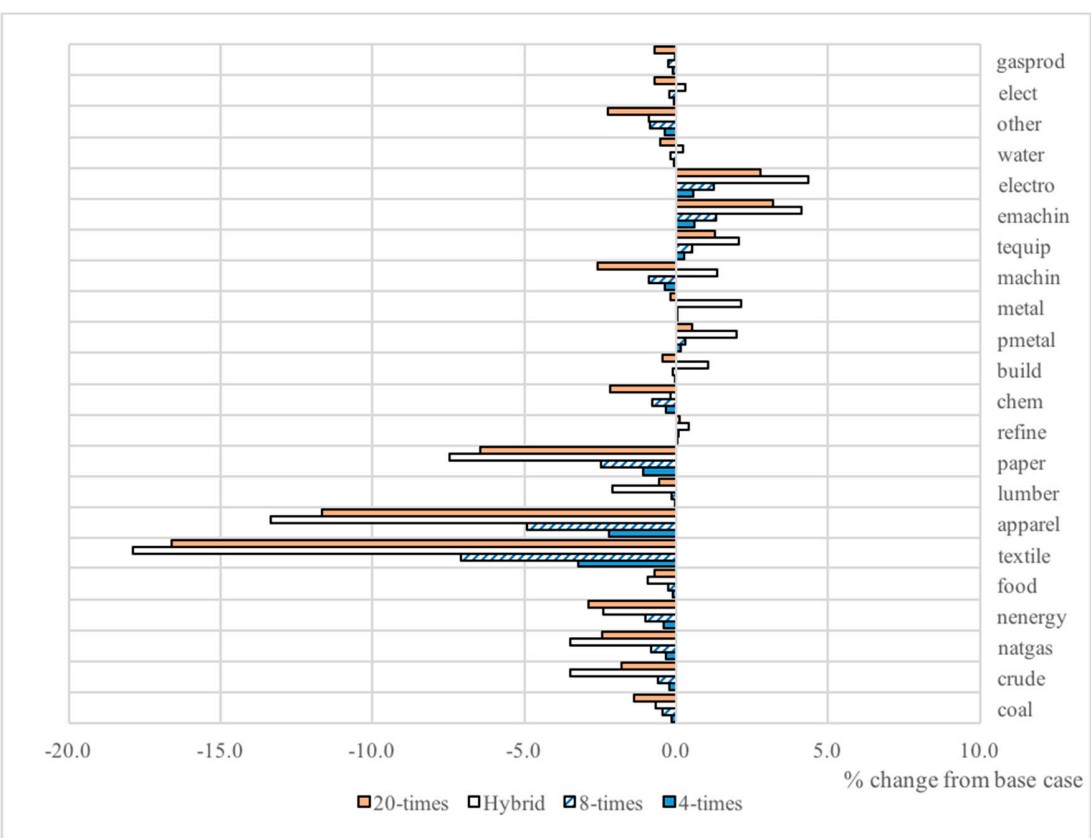

**Figure 6.** The impacts on COD discharge of 22 industrial sectors in 2030 (percent change). Negative values in the discharge change mean that the total COD discharge in the policy case is less than the total discharge in the baseline.

Compared to the 20-times policy, which taxes all industrial sectors at the same COD price, the Hybrid policy that puts the same high tax on COD-intensive sectors but a low tax on other sectors not only leads to a high COD discharge reduction that is almost equal to the reduction in the 20-times case (−2.41% versus −2.58% in 2018), but also has a bigger reduction in initial Consumption and a bigger increase in Investment. The intensity and concentration of COD discharge varies considerably among sectors (Table 1), and in the hybrid case, the discharge tax on high-concentration sectors forces them to make the biggest adjustment, while the low tax on the low-concentration, but economically important, sectors reduces the effect on the overall economy. Moreover, for the three main polluting industries (Textile goods, Wearing apparel and Paper products), the organic and non-organic pollutant (COD) in the hybrid case is cut down most among the different scenarios (Figure 7). Aggregate Consumption in the hybrid case falls by 0.12% in 2018 compared to 0.23% in the 20-times case, and Investment rises by 0.45% versus 0.41%. By 2030, the hybrid case GDP is 0.42% higher than the base case compared to 0.32% in the 20-times case.

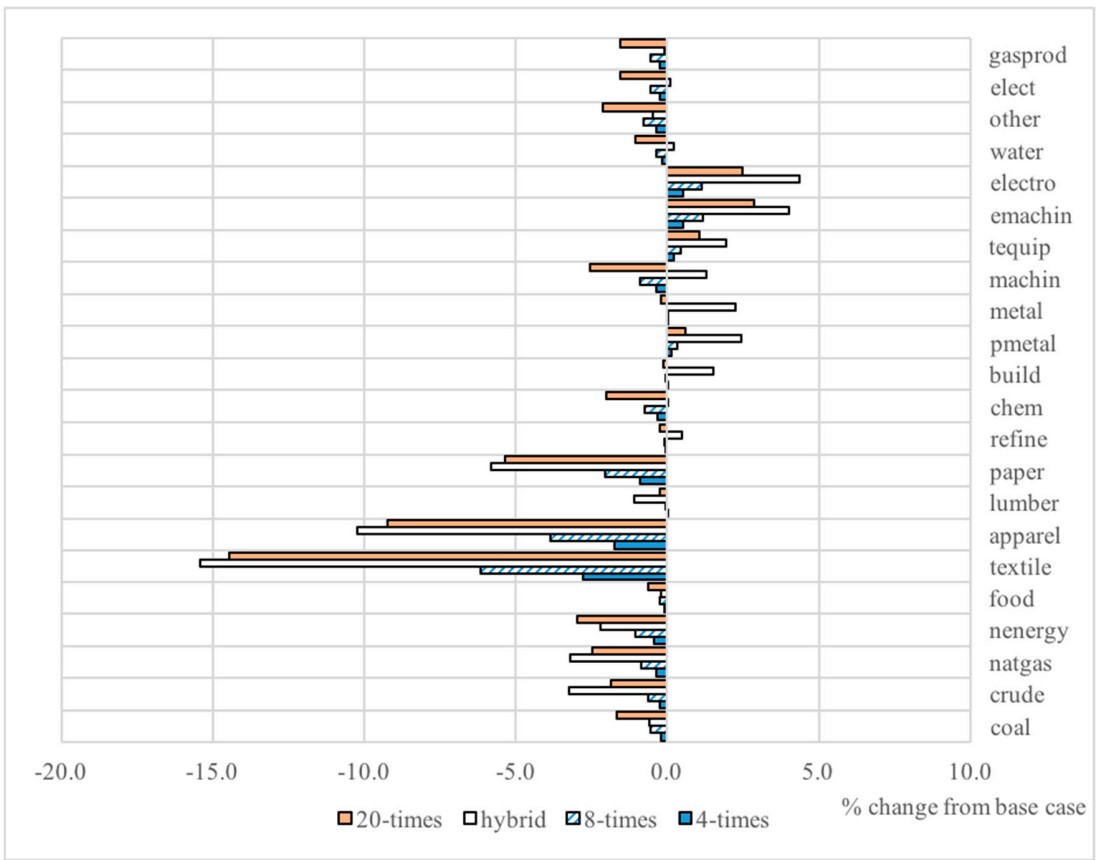

**Figure 7.** The impacts on freshwater discharge of 22 industrial sectors in 2030 (percent change). Negative values in the discharge change mean that the total COD discharge in the policy case is less than the total discharge in the baseline.

## 4. Conclusions

To sustain water resources and protect the aquatic environment, water pollution tax policies will be implemented by the Chinese government starting in 2018. This is a change from previous pollution fee policies that were not effective. Previous quality standards were also not effectively enforced. This study explores the impacts of the new market-based policies based on the amount of pollution. Economics suggest that the first best policy is to tax the emissions, which will induce firms to internalize the costs of the pollution and balance the tax with the costs of treatment. However, when the actual implementation is to impose such emission taxes only on large firms, this would not

lead to any change in the behavior of smaller producers. In this paper, we thus consider an output tax on each industry, where the level of tax depends on the COD discharge intensity. All firms have to pay this Pigovian tax, which will encourage the consumers to use less of the products that lead to the most water pollution.

We use a firm-level dataset to obtain indices of freshwater use and pollution intensity in 22 industries in 2014. We simulated a baseline growth and COD discharge with a tax equivalent to the current tax of 0.7 yuan/kg of COD out to 2030 using a multi-sector dynamic CGE model. We simulated the impacts of 3 sets of higher tax rates (2.8, 5.6 and 14 yuan/kg) and then examined a hybrid system where a 14 yuan/kg tax was placed only on the high discharge sectors, with the other sectors facing a low tax rate. We draw two major conclusions from the results:

1.  Output taxes based on water pollution rates would have a positive impact on conserving freshwater use and reducing COD discharge. Raising the tax rate on COD discharge from the current 0.7 yuan/kg by 4 times would lead to a 0.7% reduction in 2030 discharge. An increase by 20 times, to levels observed in the strictest provinces today, could lead to a 4.0% reduction in national COD discharge by 2030.
2.  If the new COD discharge tax is offset by cutting other taxes, then the loss of economic efficiency would be small. There is a small loss in initial Consumption, but a cut of taxes on enterprises would lead to higher investment and higher future GDP.

Our simulations of a hybrid system with a higher tax rate for sectors with high pollution intensities but lower rates for others show that it is worth considering—the reduction of water pollution is almost as high as that in the high tax scenario, but the initial consumption losses are smaller.

The effectiveness of any policy depends on the implementation. The old pollution fee policy was not different in theory from the pollution tax, but was not consistently implemented. The Pigovian tax on industry output may not be the first best instrument, since it does not directly encourage emission reduction, but it may be a policy that is easier to implement. Effective policies would require effective monitoring. When detailed data on the cost of water treatment becomes available, we should consider a direct tax on COD discharge in future research. This would set up a direct incentive to choose production methods and emission controls that reduce COD discharge; firms that develop lower cost technologies will then reduce emissions instead of paying the tax. Such a policy would need to take into account the pollution monitoring costs. In the output tax analyzed here, the government only needs to monitor gross output.

We have used a national model of the economy, but recognize that there are regional differences in vulnerability to pollution, in availability of water and in economic development. There are also differences in behavior between enterprises of different sizes and ownership structure. A model that takes these spatial and firm differences into account would also be an important step in future research.

**Author Contributions:** All authors made a contribution to this manuscript. X.G., J.C. and Y.H. conceived and designed the methods and collected the data; L.Y. and Y.F. analyzed the data; M.S.H., T.T. and X.G. implemented the model and analyzed the results.

**Funding:** This research was funded by National Natural Science Foundation of China under grant NO. 71461010701, the National Key Research and Development Program of China under grant NO. 2016YFE0102400, National Natural Science Foundation of China under grant NO. 71422013, Harvard Global Institute, Hang Lung Center for Real Estate, Tsinghua University Research Center for Green Economy and Sustainable Development.

**Acknowledgments:** Reviewers and Editors' comments were highly appreciated.

**Conflicts of Interest:** The authors declare no conflict of interest.

## Appendix A

Our CGE Model is from the Harvard China Project, and in particular the book entitled Clear Skies Over China: https://chinaproject.harvard.edu/clearer-skies-over-china-reconciling-air-pollution-climate-and-economic-goals. This appendix summarizes the key features of this CGE model. This is a

multi-sector model of economic growth where the main drivers of growth are population, total factor productivity growth and changes in the quality of labor and capital. It has a dynamic recursive structure, i.e., where investment is determined by a fixed savings rate, as in the Solow model. Consumption demand is driven by a translog household model that distinguishes demand by different demographic groups. We discuss the five main actors in the economy in turn—producers, households, capital owners, government and foreigners. For easy reference, Table A1 lists variables which are referred to with some frequency.

To reduce unnecessary notation, we drop the time subscript, $t$, from our equations whenever possible.

*Appendix A.1. Production*

Each of the 33 industries (Table A1) is assumed to produce its output using a constant return to scale technology. For each sector $j$, the output at time $t$, $QI_{jt}$, is expressed as:

$$QI_j = f\left(KD_j, LD_j, TD_j, A_{1j}, \ldots, A_{nj}, t\right) = f(VE, M, t) \tag{A1}$$

where $KD_j$, $LD_j$, $TD_j$, and $A_{ij}$ are capital, labor, land, and intermediate inputs, respectively. QIj denotes the quantity of industry j's output. This is to distinguish it from, QCj, the quantity of commodity j. In the actual model each industry may produce more than one commodity and each commodity may be produced by more than one industry. In the language of the input output tables, we make use of both the USE and MAKE matrices. For ease of exposition we ignore this distinction here. The *PI* and *QI* names are chosen to reflect that these are domestic industry variables, as opposed to commodities (PC) or total supply (PS), the sum of domestic output and imports.

**Table A1.** Sectors and commodities in the dynamic CGE model of China.

| | Sector | | Sector |
|---|---|---|---|
| 1 | Agriculture (Agri) | 18 | Electrical machinery (Emachin) |
| 2 | Coal mining (Coal) | 19 | Commmunication equip, computer, electronic (Electro) |
| 3 | Oil mining (Crude) | 20 | Water utilities (Water) |
| 4 | Natural gas mining (Natgas) | 21 | Other manufacturing, recycling (Other) |
| 5 | Non-energy mining (Nenergy) | 22 | Electricity, steam (Elect) |
| 6 | Food products (Food) | 23 | Gas utilities (Gasprod) |
| 7 | Textiles | 24 | Construction (Constr) |
| 8 | Apparel, leather (Apparel) | 25 | Transportation equipment (Transp) |
| 9 | Sawmills and furniture (Lumber) | 26 | Telecommunications, Software and IT (Commun) |
| 10 | Paper, printing, recording media (Paper) | 27 | Wholesale and Retail (Commerc) |
| 11 | Petroleum processing (Refine) | 28 | Hotels and Restaurants (Hotel) |
| 12 | Chemicals (Chem) | 29 | Finance |
| 13 | Nonmetal mineral products (Build) | 30 | Real estate (Realest) |
| 14 | Primary metals (Pmetal) | 31 | Business services (Busines) |
| 15 | Metal products (Metal) | 32 | Other services (Service) |
| 16 | Machinery (Machin) | 33 | Public administration (Admin) |
| 17 | Transportation equipment (Tequip) | | |

The capital stock in each industry consists of two parts—the fixed capital, $\overline{K}_j$, that is inherited from the initial period, and the market portion, $\widetilde{KD}_j$, that is rented at the market rate. The before-tax return to the owners of fixed capital in sector j is:

$$profit_j = PI_j QI_j - \widetilde{P_j^{KD}} \widetilde{KD}_j - PL_j LD_j - PT_j TD_j - \sum_i PS_i A_{ij} \tag{A2}$$

where $\widetilde{P_j^{KD}}$, $PL_j$, and $PT_j$ are market capital prices, labor prices, land prices, respectively. For each industry, given the capital stock $\overline{K_j}$ and prices, the first-order conditions from maximizing Equation (A2), subject to Equation (A1) determine the input demands.

We represent the production structure with the cost dual, expressing the output price as a function of input prices and an index of technology. The 3 primary factors and 33 intermediate inputs for each industry are determined by a nested series of constant elasticity of substitution (CES) functions taken from the GTAP model [27].

The nested structure of inputs is given in Figure A1. At the top tier, output is a function of the primary factor-energy basket (*VE*) and the non-energy intermediate input basket (*M*): $QI_j = f(VE_{j,t}, M_{j,t}, t)$. The VE basket is an aggregate of value added (*VA*) and the energy basket (*E*). Value added is a function of the 3 primary factors—capital (K), labor (L) and land (T). The constant elasticity of substitution (CES) form is used at all nodes except for non-energy intermediates where a Cobb-Douglas form is used.

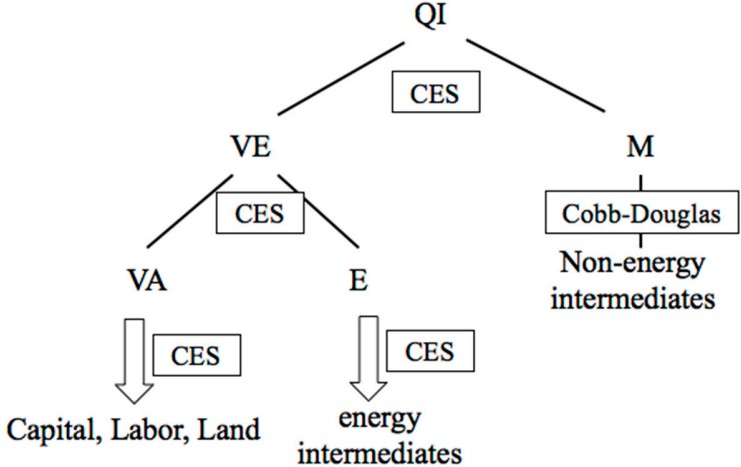

**Figure A1.** Production structure.

We denote prices of the various input by a "*P*" variables, e.g., $P^{VE}$ is the price of primary factor-energy bundle, *VE*. The top tier value equation and cost function are, respectively:

$$\begin{cases} PI_{jt}QI_{jt} = P_{jt}^{VE}VE_{jt} + PM_{jt}M_{jt} \\ PI_{jt} = \frac{\kappa_{jt}^{QI}}{g_{jt}}[\alpha_{Mjt}^{\sigma_{jt}^{QI}}PM_{jt}^{(1-\sigma_{jt}^{QI})} + (1-\alpha_{Mjt})^{\sigma_{jt}^{QI}}P_{jt}^{VE(1-\sigma_{jt}^{QI})}]^{\frac{1}{1-\sigma_{jt}^{QI}}} \end{cases} \tag{A3}$$

$g_{jt}$ is the index of technology and $\alpha_{Mjt}$ is the share of intermediate input in total costs. These $\alpha_{Mjt}$ share coefficients are allowed to change over time to reflect biases in technical change. The corresponding value equations for the primary factor-energy basket and the value-added basket are:

$$P_{jt}^{VE}VE_{jt} = P_{jt}^{VA}VA_{jt} + PE_{jt}E_{jt} \tag{A4}$$

$$P_{jt}^{VA}VA_{jt} = P_{jt}^{KD}KD_{jt} + PL_{jt}LD_{jt} + PT_{jt}TD_{jt} \tag{A5}$$

The energy basket equations give the demands for the 6 types of energy by industry *j*:

$$PE_{jt}E_{jt} = \sum_{K \in IE} PS_{kt}A_{kjt} \tag{A6}$$

The non-energy input basket is a Cobb-Douglas function of the remaining non-energy sectors (denoted *NE*; 33 − 6 = 27 components), and the corresponding equations are:

$$lnPM_{jt} = \sum_{K \in NE} \alpha^M_{kjt} lnPS_{kt} \qquad PM_{jt}M_{jt} = \sum_{K \in NE} PS_{kt}A_{kjt} \tag{A7}$$

The input shares are projected using the base year China IO-2014 (http://www.stats.gov.cn/tjsj/ndsj/2016/indexch.htm) and the US IO table of 2007 (https://www.bls.gov), supplemented by projections from the International Energy Agency [31]. The energy shares $\alpha_{Ej}$ are mostly projected to fall gradually over the next 40 years while the labor coefficient, $\alpha_{Lj}$, rises correspondingly. The coal-oil-gas-electricity components of the aggregate energy input $E_j$ also change over time.

The price to buyers of industry output includes the indirect tax on output ($t^t_i$), the externality ad-valorem tax ($t^x_i$), and the carbon tax per unit output ($t^c_i$):

$$PI^t_i = \left(1 + t^t_i + t^x_i\right)PI_i + t^c_i \tag{A8}$$

Industries versus Commodities

The model distinguishes industries from commodities, as in the official Use and Make input–output tables. Each industry may make a few commodities, and each commodity may be made by a few industries; the entry $M^{make}_{jt}$ in the Make table gives the tax-inclusive yuan value of the *i*-th commodity produced by industry *j*. The total quantity of domestic commodity is denoted $QC_i$ and its price *PC*; the sum of column *i* in the Make matrix gives the value of commodity *i* ($VQC_i$), and the sum of row *j* is the industry output value. The relation between commodity and industry output and prices are written as:

$$VQC_i = PC_iQC_i = \sum_j m^r_{jt}PI^t_jQI_j \qquad lnPC_i = \sum_j m^c_{jt}lnPI^t_j \tag{A9}$$

where $m^r_{jt}$ is the row share and $m^c_{jt}$ is the column share for the *j*-th column.

*Appendix A.2. Households*

Private consumption in this model is driven by an aggregate demand function that is derived by aggregating over different household types. Each household derives utility from the consumption of commodities, is assumed to supply labor inelastically, and owns a share of the capital stock. It also receives income transfers from the government and foreigners, and receives interest on its holdings of public debt. Aggregate private income ($Y^p$) is the sum over all households, and this income, after taxes and the payment of various non-tax fees (*FEE*), is written as:

$$Y^p = YL + DIV + G_I + G_{transfer} + R_{transfer} - FEE \tag{A10}$$

*YL* denotes aggregate labor income from supplying *LS* units of effective labor, less income taxes. Aggregate supply LS is not a simple number of working age population, but an index of work hours of different age and educational attainment groups weighted by their wages. The projected LS includes an estimate of ageing and future improvements in education. *DIV* denotes dividends from the households' share of capital income and is explained below in Equation (A17). $G_I$ and $G_{transfer}$ represent interest and transfers from the government, and $R_{transfer}$ is transfers from the rest-of-the-world. Household income is allocated between consumption ($VCC_t$) and savings. This is a Solow growth model with an exogenous savings rate that determine private savings.

Total consumption expenditures are allocated to the 33 commodities identified in the model. We do this with a demand function estimated over household consumption survey data. This consumption data is at purchaser's prices and follows the expenditure classification; these have to be linked later to the IO (Input-Output) classifications and the factory-gate prices of the IO system. At the top tier,

total expenditure is categorized as Food, Consumer Goods, Housing and Services. In the sub-tiers, these four bundles are categorized into 27 items.

Household k's indirect utility function over the four aggregates in the top tier, *V(p, M_k)*, is of a form that allows for exact aggregation:

$$lnV_k = \alpha_0 + \ln\left(\frac{p_k}{M_k}\right)' \alpha_p + \frac{1}{2}\ln\left(\frac{p_k}{M_k}\right)' B \ln\left(\frac{p_k}{M_k}\right) + \ln\left(\frac{p_k}{M_k}\right)' B_{pA} A_k \tag{A11}$$

where $M_k$ is the expenditure of household $k$, and $p_k$ is the price vector of the 4 bundles. *B* represents Bridge, which is used to link Consumption Expenditures to Input–Output accounts. Each household type has its own distinct utility function, and $A_k$ is a vector of demographic dummy variables to indicate the size of the household, presence of children, age of the head, and region. Next, the aggregate expenditures on the 4 bundles are allocated to the 27 commodities according to a nested tier structure. This is done with a linear logarithmic function that allows the shares to change over time.

*Appendix A.3. Government and Taxes*

Public revenue comes from direct taxes on capital ($t^k$), value-added taxes ($t^v$), indirect taxes on output, tariffs on imports ($t_i^r$), the externality tax ($R_{EXT}$), and other non-tax receipts:

$$Rev = \sum_j t^k(P_j^{KD}KD_j - D_j) + t^v \sum_j (P_j^{KD}KD_j + PL_jLD_j + PT_jTD_j) + \sum_j t_j^t PI_j QI_j$$
$$+ R_{EXT} + \sum_i t_i^r PM_j M_j + \sum_i t_i^c(QI_j - X_j + M_j) + FEE + R\_TWP \tag{A12}$$

where $D_j$ is the depreciation allowance and $X_j$ and $M_j$ are the exports and imports of good *i*. R_TWP is the revenue from taxes on water pollution:

$$R\_TWP = \sum_j t_j^{Ow} PI_j MQI_j \tag{A13}$$

$$t_j^{Ow} = t_j^{water} I_j^{COD} I_j^{WW} \tag{A14}$$

where $t_j^{Ow}$ is the water pollution tax rate (ad-valorem tax on output), $t_j^{water}$ is the tax per unit *COD* discharge.

Total government expenditure is the sum of commodity purchases and other payments. $G_{INV}$ is government portion of investment; $G_I$ is government net interest payments to its country; $G_{IR}$ is government net interest payments to the rest of the world; *G_transfer* is government transfer.

$$Expend = VGG + G_{INV} + G_I + G_{IR} + G\_transfer \tag{A15}$$

Government purchases of specific commodities are allocated as shares of the total value of government expenditures, *VGG*. The deficit and interest payments are set exogenously and Equation (A13) is satisfied by making the level of total nominal government expenditure on goods, *VGG*, endogenous in the base case. In simulating policy cases, we would often set the real government expenditures in the policy case equal to those in the base case.

*Appendix A.4. Capital, Investment, and the Financial System*

We model capital and investment in a simple manner. Many state-owned enterprises receive investment funds directly from the state budget and are allocated credit on favorable terms through state-owned banks. Non-state enterprises do not get such favorable treatment. We abstract from these

features and define the capital stock in each sector *j* as the sum of two parts, plan and market capital; the plan portion evolves with plan investment and depreciation:

$$K_{jt} = \overline{K_{jt}} + \widetilde{K_{jt}} ; \quad \overline{K_{jt}} = (1 - \delta)\overline{K_{jt-1}} + \psi_t^I \overline{I_{jt}} \tag{A16}$$

The rate of depreciation is $\delta$, and $\psi_t^I$ is an aggregation coefficient that converts the investment units to capital stock units (K and I are aggregates of many asset types but with different compositions). In this formulation, $\overline{K_{j0}}$ is the capital stock in sector j at the beginning of the simulation, $\overline{I_{jt}}$ is the plan investment in sector *j* at time *t*. This portion is assumed to be immobile across sectors. Over time, with depreciation and limited government investment, it will decline in importance. Each sector may also rent capital from the total stock of market capital. As discussed below, total investment in the model is determined by savings. This total, VII, is then distributed to the individual investment goods sectors through fixed shares.

After taxes, income derived from capital and land is either kept as retained earnings by the enterprises, distributed as dividends (*DIV*), or paid to foreign owners, and $B^*$ means private foreign debt.

$$\sum_j profits_j + \sum_j \widetilde{P}_j^{KD}\widetilde{K}_j + \sum_j PT_j T_j = tax(k) + RE + DIV + r(B^*) \tag{A17}$$

*Appendix A.5. The Foreign Sector*

Trade flows are modeled using the method followed in most single-country models. Imports are considered to be imperfect substitutes for domestic commodities and exports face a downward sloping demand curve. We write the total domestic supply of commodity *i* as a CES function of the domestic ($DC_i$) and imported goods ($M_i$):

$$DS_i = A_0 \left( \alpha^d DC_i^\rho + \alpha^m M_i^\rho \right)^{\frac{1}{\rho}} \tag{A18}$$

where *DC* is the quantity of domestically produced goods that are sold domestically. The price of imports to buyers is the foreign price plus tariffs, multiplied by a world relative price, *e*:

$$PM_i = e(1 + t_i^r)PM_i^* \tag{A19}$$

Domestically produced commodities (*QC*) are allocated to the domestic market and exports according to a constant elasticity of transformation (CET) function:

$$QC_{it} = k_{it}^x (\alpha_{it}^x X_{ii}^{\frac{\sigma-1}{\sigma}} + (1 - \alpha_{it}^x) DC_{ii}^{\frac{\sigma-1}{\sigma}})^{\frac{\sigma}{\sigma-1}} \tag{A20}$$

The ratio of exports to domestically sold goods depends on the domestic price (PD) relative to world prices adjusted for export subsidies. The weights and constant terms are set using base year values. The share parameters are projected exogenously to take into account the rising role of exports during 1980–2010 and a falling role in the future. The current account balance is equal to exports minus imports (valued at world prices before tariffs), less net factor payments, plus transfers.

*Appendix A.6. Markets*

The economy is in equilibrium in period t when the market prices clear the markets for the 33 commodities and the three factors. The domestic supply of commodity *i* must satisfy the total of intermediate and household consumption ($C_i$), investment ($I_i$), and government consumption ($G_i$):

$$DS_i = \sum_j A_{ij} + C_i + I_i + G_i \tag{A21}$$

For the labor market, we assume that labor is perfectly mobile across sectors, so there is one average market wage which balances supply and demand. As is standard in models of this type, we reconcile this wage with the observed spread of sectoral wages using wage distribution coefficients.

In this model without foresight, investment equals savings. There is no market where the supply of savings is equated to the demand for investment. The sum of private savings by households ($S^p$), businesses (as retained earnings: $RE$), and the government ($G_{INV}$) is equal to the total value of investment plus the budget deficit ($\Delta G$) and net foreign investment ($CA$):

$$S^p + RE + G_{INV} = VII + \Delta G + CA \tag{A22}$$

The budget deficit and current account balance are fixed exogenously in each period. The world relative price ($e$) adjusts to hold the current account balance at its exogenously determined level. The model is a constant returns-to-scale model and is homogenous in prices; that is, doubling all prices leaves the economy unchanged. We are free to choose a price normalization.

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
