# Peer review of "Industrial Water Pollution Discharge Taxes in China: A Multi-Sector Dynamic Analysis"

_water, doi:10.3390/w10121742_

Round 1
Reviewer 1 Report
The manuscript presents the application of a Computable General Equilibrium model (CGE) to water pollution control activities in China, specifically the reduction of Chemical Oxygen Demand (COD) discharged by various economic sectors to the waters of the country. While this reviewer feels that some of the underlying assumptions made by the authors could be debated (specifically the assumption about the role of the central government in industry), the paper does provide a "proof of concept" insofar as the utility of CGE modelling is concerned. Consequently, this reviewer recommends acceptance after minor modification of the manuscript. Specifically, attention is required to abbreviations and symbols used in the paper. While values are often quoted in billions of yuan, "billions" is sometimes written out in full, sometimes expressed as bil, and sometimes in other formats. This reviewer assumes that billions refers to 109. Similarly the industries are sometimes abbreviated and sometimes spelled out in full. There should be consistent usage. Otherwise, the figures, tables and references are all appropriate and the references are current. Minor editing to ensure consistent usage of terms and symbols is recommended.
Author Response
Response to Reviewer 1 Comments
The manuscript presents the application of a Computable General Equilibrium model (CGE) to water pollution control activities in China, specifically the reduction of Chemical Oxygen Demand (COD) discharged by various economic sectors to the waters of the country. While this reviewer feels that some of the underlying assumptions made by the authors could be debated (specifically the assumption about the role of the central government in industry), the paper does provide a "proof of concept" insofar as the utility of CGE modelling is concerned. Consequently, this reviewer recommends acceptance after minor modification of the manuscript.
Point 1: Specifically, attention is required to abbreviations and symbols used in the paper. While values are often quoted in billions of yuan, "billions" is sometimes written out in full, sometimes expressed as bil, and sometimes in other formats. This reviewer assumes that billions refers to 109.
Response 1: Thanks a lot. We have unified them as “billion ton” or “billion yuan”.
Point 2: Similarly the industries are sometimes abbreviated and sometimes spelled out in full. There should be consistent usage. Otherwise, the figures, tables and references are all appropriate and the references are current. Minor editing to ensure consistent usage of terms and symbols is recommended.
Response 2: Again thank you for paying attention to the details. We corrected this, and spelled out the industries’ full names in the paper except for those in figures.
Submission Date
04 November 2018
Date of this review
15 Nov 2018 01:23:20

Reviewer 2 Report
At first, I come to the conclusion that the paper is written well and the research topic has some relevance. The paper is well-structured and the line of arguments is clear and convincing from a viewpoint of an economist. Therefore, I have only a few concerns, which should be considered in my view. However, I have to note that I have to believe in the correctness of results and reliability of the used data, because I do not have access to the details of the underlying CGE model and the data. From a scientific point of view this a problem, which of course is not created by the authors.
The authors should have in mind that journal Water is a multi-disciplinary journal with a very diverse readership. For economists reading the paper does not create obstacles, but I can imagine that for readers who are not much common with economics a few more explanations are necessary. Particularly:
-The authors should explain the intuition and methodology of CGE modelling and provide its theoretical background, so that a non-economist has a chance to understand what it that is and what it does. That does not mean that the authors have to explain the used model in detail, but to provide a general description.
-Further to justify the use of a CGE approach by stating that others have done it also, is not really convincing. A better foundation for the authors’ choice should be provided.
-In a similar line, the authors should also consider the limitations of this approach regarding their results.
-Explain in a footnote what a SAM is, where I am wondering why the respective data regarding the environment can be found in a SAM and not in a NAMEA (national accounting matric including environmental accounts).
-According to the authors the tax is imposed on industry output and not directly on emissions (page 7). This has to be discussed, because this raises important questions (e.g. the implicitly assumed linear relationship between output and COD discharges.)
- How to justify the assumption that government consumption is been fixed for the next 12 years (page 8), although the economy is assumed to grow by 5.2% pa? Usually I would expect that government consumption grows at the same rate as the economy.
-The reference style in the text is not consistent, sometimes per endnote (what is awful for the reader), sometimes endnote number plus author and year) (page 6) (Wu at al (2015)), but sometimes only like Fan et al (page 3). This has to be made consistent.
Minor concerns:
What is Grade III? (page 2)
Correct: As has been emphasized by many papers…(page 7). Papers do not emphasize anything, only authors.
What means GTAP model (page 7)?
Author Response
Response to Reviewer 2 Comments
At first, I come to the conclusion that the paper is written well and the research topic has some relevance. The paper is well-structured and the line of arguments is clear and convincing from a viewpoint of an economist. Therefore, I have only a few concerns, which should be considered in my view. However, I have to note that I have to believe in the correctness of results and reliability of the used data, because I do not have access to the details of the underlying CGE model and the data. From a scientific point of view this a problem, which of course is not created by the authors.
The authors should have in mind that journal Water is a multi-disciplinary journal with a very diverse readership. For economists reading the paper does not create obstacles, but I can imagine that for readers who are not much common with economics a few more explanations are necessary. Particularly:
Point 1: The authors should explain the intuition and methodology of CGE modelling and provide its theoretical background, so that a non-economist has a chance to understand what it that is and what it does. That does not mean that the authors have to explain the used model in detail, but to provide a general description.
Response 1: Thanks. This is a very valid point. Now we have added the following to explain CGE model for a non-economist. (Line 103-116)
“We analyze these policies using a class of models in economics known as Computable General Equilibrium (CGE) models. Over the past 25 years, CGE models have become a standard tool of empirical economic analysis and have been widely applied to assess the impact of environmental policies, such as those reviewed in [15]. CGE models account for inter-industry transactions, e.g., how much electricity is bought by the steel industry, how much steel is bought by vehicle manufacturers, and how many vehicles are bought by consumers. On the supply side the there are production functions that give the inputs required to produce a unit of each industry’s output as a function of input prices and gives information on the pollution emissions by that industry. On the demand side there are consumption functions that describe how households allocate their total spending to the different goods based on incomes and prices. These models are “equilibrium” in that they solve for prices that balances supply and demand for each good and for capital and labor inputs; they are “general” in that they cover the entire economy. In the CGE model, a new policy such as a COD tax would change costs and thus prices of different goods to different extents. The new set of relative prices would change consumption behavior and thus lead to a new vector of industry output.”
Point 2: Further to justify the use of a CGE approach by stating that others have done it also, is not really convincing. A better foundation for the authors’ choice should be provided.
Response 2: We have added further text in Line 104-106, and line 117-119:
“Over the past 25 years, computable general equilibrium (CGE) models have become a standard tool of empirical economic analysis. In recent years, improvements in model specification, data availability, and computer technology have improved the payoffs and reduced the costs of policy analysis based on CGE models, paving the way for their widespread use by policy analysts throughout the world.”
Point 3: In a similar line, the authors should also consider the limitations of this approach regarding their results.
Response 3: We agree with that. In fact, we did discuss the limitations in the Conclusion section. Now we have explicitly stated the limitation in Line 134-135:
“These studies of industrial wastewater applied static models to simulate the economic and environmental impacts and do not discuss the dynamic aspects of policies.”
Point 4: Explain in a footnote what a SAM is, where I am wondering why the respective data regarding the environment can be found in a SAM and not in a NAMEA (national accounting matric including environmental accounts).
Response 4: We are now explaining what a SAM is in more detail in the main text in section 2.1. The typical SAM traces the monetary transactions (sales, transfers, taxes) between industries, households, capital owners, government and rest-of-the-world. We supplement that typical SAM with our emission accounts in the same way that National Accounts are supplemented in NAMEA.
We have described more details in Line 143-154. We deleted the original Footnote 4 (on SAM) moved part of information to the main text.
Point 5: According to the authors the tax is imposed on industry output and not directly on emissions (page 7). This has to be discussed, because this raises important questions (e.g. the implicitly assumed linear relationship between output and COD discharges.)
Response 5: Indeed you highlighted an important point. We have pointed that they are Pigovian taxes (line 82-88). And we further explained this at line 430-438.
“We focus on taxes on industry output where the tax rate is based on the amount of pollution generated in producing that output, what economists call Pigovian taxes. The purpose of this tax is to inject price signals into the economic system so that consumers take into account the environmental damages caused by that good when deciding how much to buy [13]. Governments may also consider a direct tax on pollution emissions such as COD discharge; such a tax is efficient but may be difficult to apply to all emissions. Under the current system the pollution tax only applied to large enterprises”
“The Pigovian tax on industry output may not be the first best instrument since it does not directly encourage emission reduction, but it may be a policy that is easier to implement. Effective policies would require effective monitoring. When detailed data on the cost of water treatment becomes available we should consider a direct tax on COD discharge in future research. This would set up a direct incentive to choose production methods and emission controls that reduce COD discharge; firms that develop lower cost technologies will then reduce emissions instead of paying the tax. Such a policy would need to take into account the pollution monitoring costs. In the output tax analyzed here, the government only needs to monitor gross output.”
Point 6: How to justify the assumption that government consumption is been fixed for the next 12 years (page 8), although the economy is assumed to grow by 5.2% pa? Usually I would expect that government consumption grows at the same rate as the economy.
Response 6: Sorry that we were not clear in our use of the term “at the baseline levels”, and have clarified it. We do not mean that government expenditures are fixed at base year levels for all future years. We mean that in the policy case, government real purchases in a given year is required to be the same as the purchases in the base case. In the model government collects taxes, purchases commodities, redistributes resources and borrow an amount equal to the deficit. In simulating policy cases, we set the real government expenditures in the policy case equal to those in the base case in each year. To do this, we implement a revenue-neutral change by using the new revenues raised by higher water taxes to cut other existing taxes.
Point 7: The reference style in the text is not consistent, sometimes per endnote (what is awful for the reader), sometimes endnote number plus author and year) (page 6) (Wu at al (2015)), but sometimes only like Fan et al (page 3). This has to be made consistent.
Response 7: We apologize for the inconsistencies. This has been corrected now.
Minor concerns:
Point 8: What is Grade III? (page 2)
Response 8: we added an explanation of Grade III (footnote 1 in page 2).
Point 9: As has been emphasized by many papers…(page 7). Papers do not emphasize anything, only authors.
Response 9: we corrected this.
Point 10: What means GTAP model (page 7)?
Response 10: We added a brief explanation of Global Trade Analysis Project (GTAP) in Footnote 12 (page 8).
Submission Date
04 November 2018
Date of this review
08 Nov 2018 09:28:30

Reviewer 3 Report
First, the authors should clarify the contributions of the paper, ideally up front in the introduction. You can talk about the following contributions: What insights can you provide based on your finding? Do they push forward our understanding? What should we do with your research? Do you have any suggestions to improve the current regulation or practice? Adding the above discussion and extend your literature review may help you make more contributions and position your contributions better.
The policy seems to have different effects on firms of different sizes. You need to test it or at least discuss and rationalize the use of firm size measure, since it is the key variable in the literature. See Dang, et al. 2018. Measuring Firm Size in Empirical Corporate Finance. Journal of Banking & Finance, 86:159-176. After all it is the most significant variable in most studies in this area. Need to explain.
In conclusion, I would like to thank the authors for a very interesting and potentially important paper, and hope these comments and suggestions can help further their study.
Author Response
Response to Reviewer 3 Comments
Comments and Suggestions for Authors
Point 1: First, the authors should clarify the contributions of the paper, ideally up front in the introduction. You can talk about the following contributions: What insights can you provide based on your finding? Do they push forward our understanding? What should we do with your research? Do you have any suggestions to improve the current regulation or practice? Adding the above discussion and extend your literature review may help you make more contributions and position your contributions better.
Response 1: We thank the reviewer for emphasizing the need to be clear in the Introduction, as did another reviewer. We have expanded the Introduction section to: (i) explain the policy simulated in a clearer way to all readers including those who are not economists; (ii) summarize the results and insights; (iii) explain the CGE modelling approach in more general language. We have also rewritten the Conclusion to summarize the results better and compare our results to the literature. We now discuss the limitations of our method and suggest useful future research efforts.
Point 2: The policy seems to have different effects on firms of different sizes. You need to test it or at least discuss and rationalize the use of firm size measure, since it is the key variable in the literature. See Dang, et al. 2018. Measuring Firm Size in Empirical Corporate Finance. Journal of Banking & Finance, 86:159-176. After all it is the most significant variable in most studies in this area. Need to explain.
Response 2: We thank the reviewer for emphasizing this. We added a paragraph at the end of section 2.1 to discuss this. We recognize the importance of these differences but note the extra research that will be required. We added a sentence in the Conclusion that recognize this important future step.
In conclusion, I would like to thank the authors for a very interesting and potentially important paper, and hope these comments and suggestions can help further their study.
Submission Date
04 November 2018
Date of this review
06 Nov 2018 18:42:57

Round 2
Reviewer 3 Report
much improved